# *TFOFinder*: Python program for identifying purine-only double-stranded stretches in the predicted secondary structure(s) of RNA targets

**Atara Neugroschl[1]¤, Irina E. Catrina[2]***

**1** Department of Chemistry and Biochemistry, Stern College for Women, Yeshiva University, New York, New York, United States of America, **2** Department of Chemistry and Biochemistry, Yeshiva College, Yeshiva University, New York, New York, United States of America

¤ Current address: Tri-Institutional Ph.D. Program in Chemical Biology, New York, New York, United States of America

* irina.catrina@yu.edu

**Data Availability Statement:** The *TFOFinder* program is freely available on GitHub: https://github.com/icatrina/TFOFinder.

## Abstract

Nucleic acid probes are valuable tools in biology and chemistry and are indispensable for PCR amplification of DNA, RNA quantification and visualization, and downregulation of gene expression. Recently, triplex-forming oligonucleotides (TFO) have received increased attention due to their improved selectivity and sensitivity in recognizing purine-rich double-stranded RNA regions at physiological pH by incorporating backbone and base modifications. For example, triplex-forming peptide nucleic acid (PNA) oligomers have been used for imaging a structured RNA in cells and inhibiting influenza A replication. Although a handful of programs are available to identify triplex target sites (TTS) in DNA, none are available that find such regions in structured RNAs. Here, we describe *TFOFinder*, a Python program that facilitates the identification of intramolecular purine-only RNA duplexes that are amenable to forming parallel triple helices (pyrimidine/purine/pyrimidine) and the design of the corresponding TFO(s). We performed genome- and transcriptome-wide analyses of TTS in *Drosophila melanogaster* and found that only 0.3% (123) of total unique transcripts (35,642) show the potential of forming 12-purine long triplex forming sites that contain at least one guanine. Using minimization algorithms, we predicted the secondary structure(s) of these transcripts, and using *TFOFinder*, we found that 97 (79%) of the identified 123 transcripts are predicted to fold to form at least one TTS for parallel triple helix formation. The number of transcripts with potential purine TTS increases when the strict search conditions are relaxed by decreasing the length of the probe or by allowing up to two pyrimidine inversions or 1-nucleotide bulge in the target site. These results are encouraging for the use of modified triplex forming probes for live imaging of endogenous structured RNA targets, such as pre-miRNAs, and inhibition of target-specific translation and viral replication.

**Funding:** This work was funded in part by the Yeshiva University Start-up Fund (IEC) and 2023-2024 Yeshiva University Faculty Research Fund (IEC). The funders had no role in the study design, data collection and analysis, decision to publish, or preparation of the manuscript.

**Competing interests:** The authors have declared that no competing interests exist.

## Author summary

Nucleic acid molecules are most often encountered in living organisms as double-stranded (DNA) or single-stranded (RNA). However, when meeting certain sequence requirements, they can also form complex structures in which three (triplex) of four (quadruplex) strands will interact. Important biological roles were reported for short intramolecular RNA triplexes and more recently it was shown that noncoding RNAs can control gene expression via intermolecular triplex formation with double-stranded DNA. Current algorithms identify double-stranded DNA regions, as well as single-stranded RNA regions that can form a triplex, but no programs are available to identify such regions in a structured RNA. We wrote *TFOFinder*, a Python program to design probes that are predicted to form intermolecular triplexes with structured regions of a given RNA target. These probes can be used for imaging structured RNAs in physiological conditions or for target-specific translation inhibition. We first analyze the fruit fly transcriptome for RNAs that show the potential to form triplexes and predict the secondary structure of all hits. Using our program, we take into consideration the structure of each target and find that most of these hits are predicted to contain regions amenable to forming triplexes.

## Introduction

In 1957, four years after Watson and Crick published the structure of double-stranded DNA, Felsenfeld, Davies, and Rich reported the characterization of poly(A)/poly(U) triple helix formation [1]. Since then, it has been revealed that DNA and RNA triple helices have important biological roles in catalysis, regulation of gene expression, and RNA protection from degradation (reviewed in [2]).

When meeting certain requirements, nucleic acids can form triple or quadruple helices. The latter is formed by G-rich sequences and recent studies revealed quadruplex selective recognition for *in vivo* analysis of human telomeric G-quadruplex formation [3]. Natural intramolecular triple helices form for nucleic acid sequences rich in consecutive purine (R) and pyrimidine (Y) stretches and were proposed to control gene expression by inhibiting transcription or preventing the binding of other factors [4]. Intermolecular triple helices are promising tools for artificial control of gene expression and as therapeutic approaches to address various human diseases [5–7], which form when a third strand interacts with a canonical duplex via Hoogsteen base pairs (bp) (Fig 1; reviewed in [2]). The third strand can bind to the major or minor groove of a duplex; however, the minor groove triplex is unstable. In addition, depending on sequence composition, the third strand can bind in a parallel or antiparallel orientation to form Y•R:Y and R•R:Y triple helices, respectively. Where "•" and ":" denote Hoogsteen and Watson-Crick hydrogen-bonding, respectively. Triplex-forming oligonucleotides (TFO) can have a DNA or RNA backbone, and when they have a length of at least 10–12 nucleotides (nt), triplex formation can be characterized with common assays, such as native gel electrophoresis [8]. With an unmodified TFO (DNA or RNA), triplex formation involves the interaction between three strands all with a negatively charged backbone, which leads to electrostatic repulsion and a very slow association of the third strand. However, once formed, parallel triple helices are very stable with half-lives of days. The peptide nucleic acid (PNA) backbone modification has been employed to eliminate this unfavorable interaction, which resulted in high TFO binding specificity and sensitivity, and with a greater mismatch discrimination as compared to using DNA or RNA TFOs [9–11]. Triplex formation can further be favored and stabilized by employing base modifications [11–17].

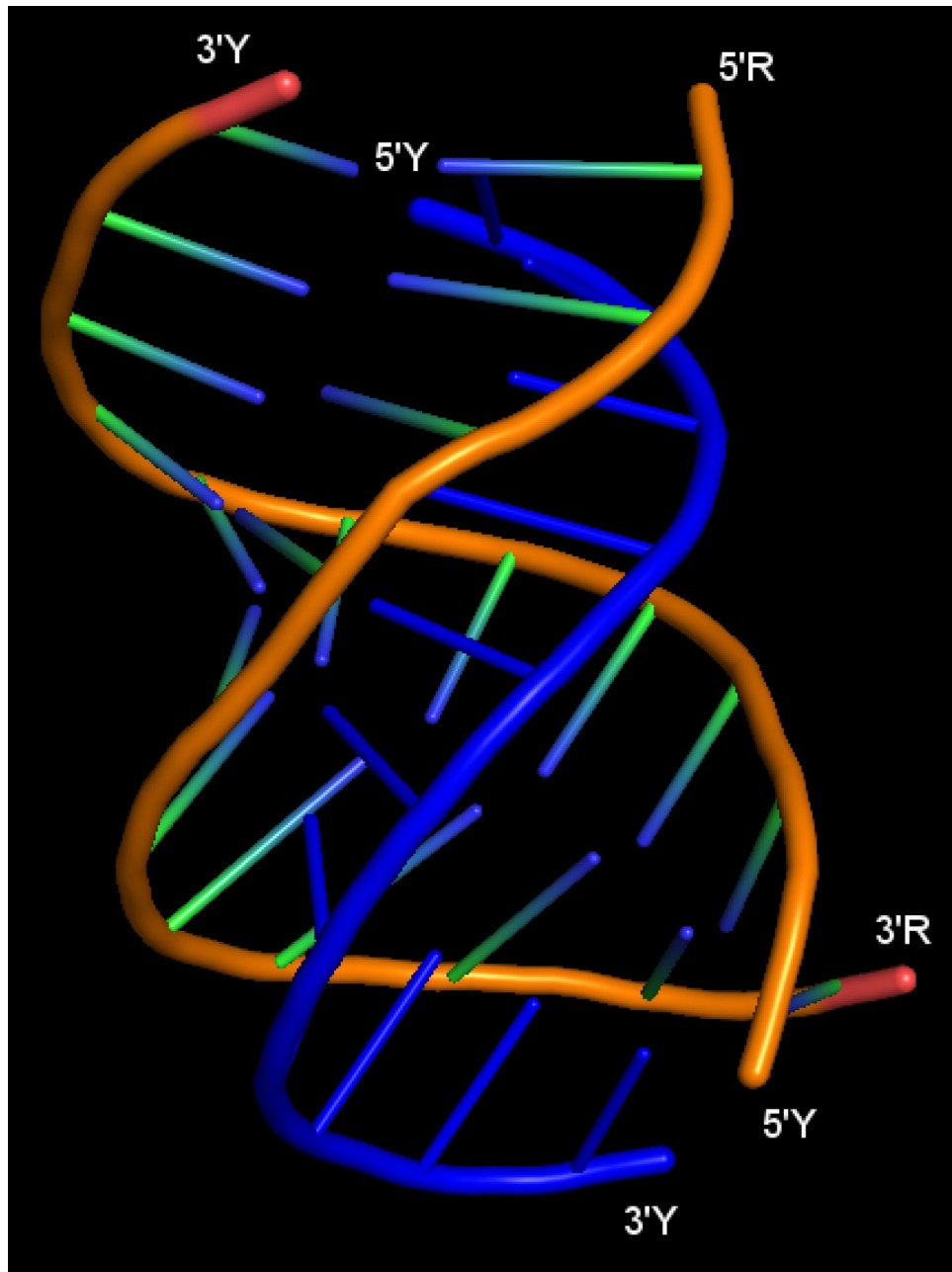

**Fig 1. Structure of an intramolecular Y•R:Y triple helix formed with an 11-nt long TTS, as determined using X-ray crystallography.** The strands forming the R:Y Watson-Crick duplex are shown in orange, and the triple helix forming Y strand is shown in blue. R = purine, Y = pyrimidine, "•" = Hoogsteen H-bonding, ":" = Watson-Crick H-bonding. Structure adapted from PDB ID: 6SVS [18] using the PyMOL Molecular Graphics System, version 2.3.2 (Schrödinger, LLC).

Endogenous DNA and RNA triple helices have important biological roles; RNA splicing (RNA•RNA:RNA) and telomere synthesis (RNA•DNA:DNA) involve the formation of short triple helices [19, 20]. In the first example, the backbone phosphates bind metal ions needed for splicing, and in the second example, triplex formation is required for catalysis. Triple helices are also involved in gene expression regulation by mediating ligand binding for metabolite-sensing

riboswitches in bacteria and facilitate RNA protection from degradation [21–26]. Exogenous RNA triple helices have great potential for application in imaging of endogenous RNAs, target-specific inhibition of translation, and inhibition of pre-miRNA processing.

The use of unmodified TFOs (DNA or RNA) is limited in general by the formation of intermolecular structures or motifs (I-motif and G-quadruplex) or duplex-formation with single-stranded regions of target and non-target RNAs. Important advances have been made in identifying backbone and base modifications to enhance TFO selectivity. These are greatly expanding TFO applications to imaging and studies of gene expression regulation. PNA•RNA:RNA triple helix formation was shown to efficiently inhibit viral replication of influenza A (IAV) [27].

Although TFOs show great promise for applications in biology and medicine, there are also a few aspects that still need to be improved:

1. Cellular, cytoplasmic, and nuclear, delivery of TFOs; efficient oligonucleotide delivery is currently achieved using various delivery agents (*e.g.*, polyamines, liposomes) and/or electroporation methods, depending on the specimen and delivery site of interest. Recently, modified oligomers showed superior cellular uptake without the use of carriers [27, 28].

2. Solubility of PNA-derived TFOs; exchanging the negatively charged phosphate diester for an uncharged peptide backbone coupled with the hydrophobicity of the nitrogenous bases can yield PNA oligomers with reduced water solubility. This is addressed by the addition of up to three positively charged amino acid residues, usually lysine, at the N- or C-terminus of the TFO.

3. TFO design for RNA targets; TFO design for double-stranded DNA targets is straightforward, one only needs to search the target DNA sequence for purine stretches with the length of interest. The *Triplexator* application was reported to predict short (< 30-bp) double-stranded DNA binding sites for a given RNA sequence [29]. *LongTarget* finds longer DNA TTS, the *Triplex Domain Finder* application detects DNA-binding domains in long non-coding RNAs, and the *Triplex* from the *R/Bioconductor* suite predicts the formation of eight types of intramolecular triplexes within a given nucleic acid sequence [30–32]. However, to our knowledge, there are no applications that facilitate the design of TFOs for structured RNA targets containing R:Y duplex regions, which can form intermolecular triplexes.

DNA and RNA triple helices have been extensively analyzed via optical melting experiments, circular dichroism, FRET (<u>F</u>luorescence/<u>F</u>örster <u>R</u>esonance <u>E</u>nergy <u>T</u>ransfer), and other techniques. Of particular interest are RNA•DNA:DNA and RNA•RNA:RNA triple helices, as they have essential biological roles, such as telomere synthesis where they ensure proper pseudoknot folding, catalysis without direct association with the active site, and recruiting divalent metal ions for splicing (reviewed in [2]). Efficient triple helix formation with a TFO containing an unmodified DNA/RNA backbone requires at least 10-bp long purine rich TTS and a mildly acidic pH to protonate cytosines such that they can participate in Hoogsteen base pairing. TTS hairpin models with purine-rich stems and random loop sequence are commonly used to analyzed TFO properties in solution (Fig 2A).

Here, we describe *TFOFinder*, an open-source Python program to design parallel pyrimidine TFOs recognizing purine-only double-stranded regions in any RNA target of interest (Y•R:Y) (Fig 2B). We used *RNAMotif* and *TFOFinder* to determine the prevalence of potential DNA, and RNA target sites in the *Drosophila melanogaster* genome (*version 6.48*) and transcriptome (*version 6.38*), respectively [33]. *RNAMotif* is a valuable and flexible tool that uses descriptor files to search for a user-defined primary or secondary structure "motif" within a given file containing one or more sequences in the FASTA format [33]. The *TFOFinder*

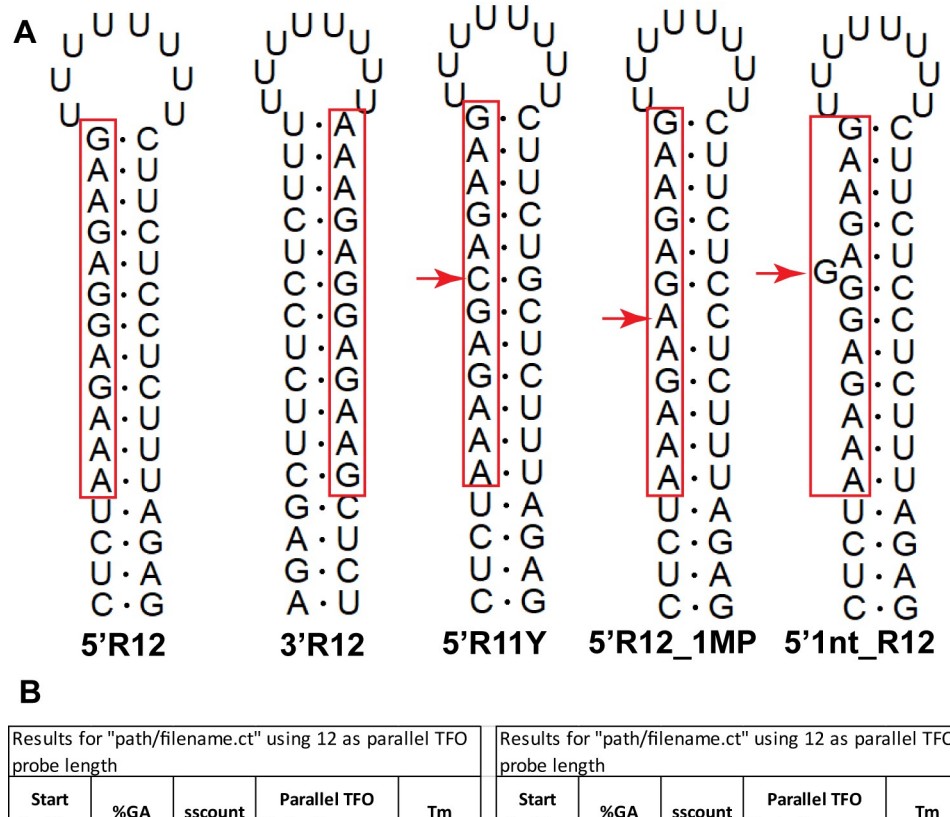

**Fig 2. Model RNA hairpins illustrating examples of ideal and interrupted 12-bp long TTS.** (A) The purine stretch (red box) can be positioned on the 5' (5'R12) or 3' (3'R12) side of the hairpin duplex, and these two TTS are readily identified by *TFOFinder*. The remaining three TTS are not reported by *TFOFinder* and can only form stable triplexes with a modified TFO. 5'R11Y = the purine region is positioned on the 5' side of the duplex and it is interrupted by a pyrimidine inversion (red arrow). 5'R12_1MP = the purine region is positioned on the 5' side of the duplex and it is interrupted by a mispair (red arrow). 5'1nt_R12 = the purine region is positioned on the 5' side of the duplex and it is interrupted by a 1-nt bulge (red arrow). (B) The *TFOFinder* output for the first two TTS RNA hairpin examples, 5'R12 and 3'R12.

program takes into consideration the predicted secondary structure(s) of an RNA target of interest and designs the corresponding TFO probe(s), features that are not implemented in *RNAMotif*. However, when large-scale transcriptome-wide studies are performed, *RNAMotif* is an invaluable tool for first identifying RNA sequences that show the potential to form purine duplexes. These results can be further analyzed using *TFOFinder*; to include structure information obtained using freely available RNA folding software (*e.g.*, *mfold* [34]; reviewed in [35]) and design TFO probes.

We show that our program facilitates the identification within any RNA target of duplex regions amenable to forming a parallel Y•R:Y triplex, and the design of the corresponding short TFO probes (4-30-nt). These TFO probes can be used for specific inhibition of translation and imaging of structured RNAs containing purine-rich sequences in non-denaturing conditions.

## Results and discussion

TFO probes have already found important applications in the imaging of cellular RNAs and nucleic acid function modulation and assays [27, 36–39]. Here, we explored the feasibility of

extending the application and versatility of TFOs by performing a transcriptome- and genome-wide analysis in *D. melanogaster* to identify all RNA and DNA stretches that are amenable to triple helix formation. Moreover, we tested our program by designing TFO probes for a previously reported RNA target, the vRNA8 of influenza A, which encodes two essential viral proteins, NEP and NS1 [27, 40, 41]. TFO probes designed using *TFOFinder* are promising tools for *in vivo* imaging of structured RNA targets (*e.g.*, pre-miRNAs), determining *in vivo* folding of endogenous RNA targets, target-specific inhibition of translation, and others.

To identify continuous single-stranded stretches of 12 purines, we searched the fruit fly transcriptome using *RNAMotif*, a program that finds user-defined sequences or potential structural motifs in a given nucleic acid target sequence without information about the target's secondary structure [33]. We counted adenine (A)-only stretches separately from guanine (G)-containing ones and identified all hits corresponding to unique transcripts. We then searched the sequence of the transcripts containing these hits for a complementary match, or a match containing G-U wobble pair(s), or with one mispair.

## *Drosophila melanogaster* genome survey

Both strands of the DNA genome were searched for R12 stretches (containing at least one G), which were identified and counted for defined DNA regions (Table 1). These stretches were found in more than 50% of targets for gene sequences. The largest number of hits were obtained for intronic regions (437,487), mapped to 23.72% of total unique intronic targets. tRNAs and miRNAs contained the least number of R12 sequences, mapped to only 0.96% (3) and 1.74% (13) of total tRNA and miRNA unique targets, respectively. However, not all R12 hits listed in Table 1 are unique, as the exon, UTRs, gene, and mRNA sequences present significant overlap.

## *Drosophila melanogaster* transcriptome survey

While triple helix formation with a DNA/RNA TFO requires the presence of a continuous stretch of purines in the target, it has been shown that triplexes can be formed with TTS containing one or two pyrimidine inversions (Fig 2A, 5'R11Y) when a modified TFO is employed. Therefore, we determined whether allowing pyrimidine inversions would significantly increase the number of transcript hits. We analyzed the full *D. melanogaster* transcriptome by beginning with a strict search (R12—all purines and not all As), which we gradually relaxed to

**Table 1. Results for the *D. melanogaster* genome (*version 6.48*) for R12.**

| Target | Total unique targets^ | # Unique DNA targets with R12* | # R12 on both target strands |
|---|---|---|---|
| mRNA | 30,799 | 7,501 | 152,443 |
| gene | 17,902 | 10,122 | 293,167 |
| exon | 85,590 | 16,086 | 84,606 |
| ncRNA | 3,053 | 1,092 | 8,894 |
| intron | 72,062 | 17,095 | 437,487 |
| intergenic | 12,347 | 4,707 | 113,125 |
| 3'UTR | 30,285 | 2,762 | 43,361 |
| 5' UTR | 30,184 | 2,549 | 35,573 |
| miRNA | 747 | 13 | 100 |
| tRNA | 312 | 3 | 5 |

^ counted using MD5 values; * counted using unique gene IDs (FBgn#), except for exons, introns, and intergenic regions, for which the MD5 value was used.

**Table 2. Results for the survey of *D. melanogaster* transcriptome (*version 6.38*) for the indicated purine-rich sequences.**

| Sequence | Total Single-stranded hits | Single-stranded: # unique transcripts | Single-stranded: # unique genes | Total Double-stranded hits | Double-stranded: # unique transcripts | Double-stranded: # unique genes |
|---|---|---|---|---|---|---|
| A12 | 8,707 | 2,300 (6.5%) | 1,031 (5.8%) | 2,453 | 217 (0.6%) | 105 (0.6%) |
| R12 | 128,139 | 16,689 (46.8%) | 7,076 (39.6%) | 494 | 123 (0.3%) | 54 (0.3%) |
| R12_GU | n/a | n/a | n/a | 31,213 | 1,506 (4.2%) | 620 (3.5%) |
| R12_1MP | n/a | n/a | n/a | 5,046 | 811 (2.3%) | 351 (2.0%) |
| R11Y | 588,205 | 30,793 (86.4%) | 14,601 (81.7%) | 813 | 391 (1.1%) | 178 (1.0%) |
| R10Y2 strict | 1,402,460 | 33,935 (95.2%) | 16,733 (93.6%) | 438 | 317 (0.9%) | 138 (0.8%) |
| R10Y2 relaxed | 2,663,079 | 34,259 (96.1%) | 16,993 (95.0%) | 890 | 606 (1.7%) | 269 (1.5%) |

A12 = 12 consecutive adenines; R12 = 12 consecutive purines, containing at least one guanine; R12_GU = R12 duplex that may contain one or more G-U wobble base pairs (including R12 hits); R12_1MP = R12 duplex that may contain one mispair/mismatch (including one G-U as mispair and R12 hits); R11Y = 12 consecutive nucleotides composed of eleven purines (A11 or AiGj with i+j = 11) and one internal pyrimidine; R10Y2 = 12 consecutive nucleotides composed of ten purines (A10 or AiGj with i+j = 10) and two pyrimidines; strict = the two Ys are not next to each other and not at the ends; relaxed = two Ys anywhere (including R10Y2 strict hits). Total number of transcripts = 35,642; Total number of genes = 17,878.

allow for G-U pairing, or one mispair (Fig 2A, 5'R12_1MP), or up to two pyrimidine inversions in accordance with previously reported triple helix formation rules and restrictions (Table 2 and Fig 3) [12, 14, 42–44]. For the strictest search, for R12 sequences, we identified all 12-nt stretches of purines that had at least one G and found that 123 unique transcripts (0.3% of the total 35,642 transcripts) also contained at least one corresponding complementary sequence needed to form an R12 TTS, which were encoded within 54 unique genes (0.3% of the total 17,878 genes; Table 2, R12; S1 Table). When G-U paring was allowed, we identified

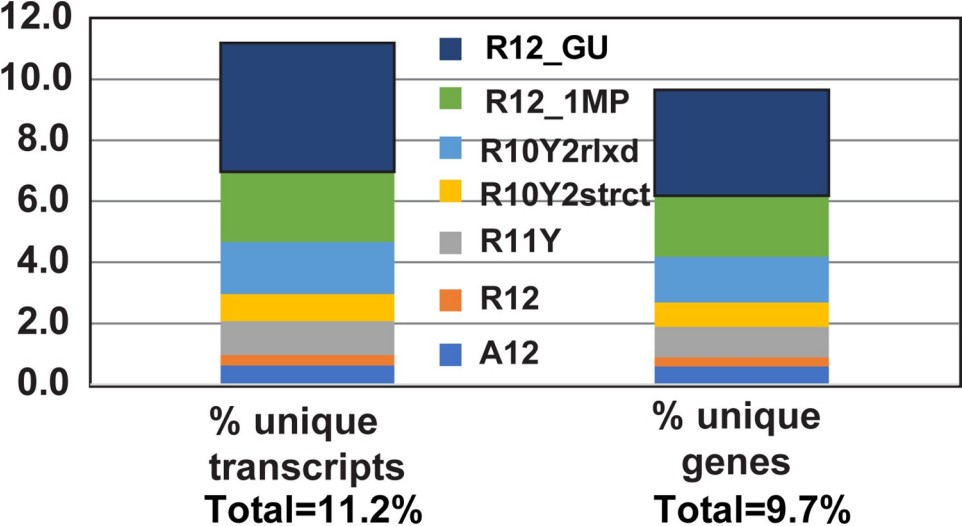

**Fig 3. The percentage of unique transcripts and corresponding genes containing the indicated TTS types, as obtained from the transcriptome (*version 6.38*) analysis.** A12 = 12 consecutive adenines; R12 = 12 consecutive purines, containing at least one guanine; R12_GU = R12 duplex that may contain G-U base pair(s); R12_1MP = R12 duplex that may contain one mispair/mismatch; R11Y = 12 consecutive nucleotides composed of eleven purines (A11 or AiGj with i+j = 11) and one internal pyrimidine; R10Y2 = 12 consecutive nucleotides composed of ten purines (A10 or AiGj with i+j = 10) and two pyrimidines; strict (strct) = two Ys not next to each other and not at the ends; relaxed (rlxd) = two Ys anywhere. Total number of transcripts = 35,642; Total number of genes = 17,878.

1,506 (4.2%) unique transcripts mapped to 620 (3.5%) unique genes containing complementary sequences with the potential of forming 12-bp long purine duplexes (Table 2, R12_GU). When one mispair was allowed, there were 811 (2.3%) unique transcripts mapped to 351 (2.0%) unique genes containing complementary sequences with the potential of forming interrupted 12-bp long purine duplexes (Table 2, R12_1MP). When we relaxed the conditions to allow for one internal pyrimidine inversion (Table 2, R11Y) and eliminated the requirement for a G, 391 (1.1%) unique transcripts were identified, corresponding to 178 (1.0%) unique genes (Table 2, R11Y). Finally, we also allowed for two pyrimidine inversions (R10Y2). First, we restricted the position of the inversions to the middle of the TTS, and not consecutive. With these search restrictions we found 317 (0.9%) unique transcripts, corresponding to 138 (0.8%) unique genes (Table 2, R10Y2 strict). Second, we relaxed the R10Y2 strict search to allow the two non-consecutive, internal pyrimidine inversions to be consecutive and/or terminal. Under these conditions, we discovered 606 (1.7%) unique transcripts mapped to 269 (1.5%) unique genes (Table 2, R10Y2 relaxed).

Using the PANTHER Classification System, we performed a gene ontology enrichment analysis for molecular functions and biological processes for the 54 unique genes from which the 123 transcripts with potential TTS are expressed [45]. This analysis identified 13 (28.3%) genes with binding as molecular function, 14 (30.4%) and 13 (28.3%) genes involved in biological regulation and metabolic processes, respectively. However, 50% or more of these genes were not assigned to any PANTHER category.

### *TFOFinder*

The *TFOFinder* program, to our knowledge, is the first to search within the predicted secondary structure(s) of an RNA target of interest for double-stranded fragments of a user-defined length (4-30-nt) that are composed of consecutive purines (*i.e.*, R12, R12_GU, and A12). The *TFOFinder*'s flow chart shows the main steps of the program (Fig 4). The program identifies purine-only regions that are double-stranded and can include G-U wobble pairs, within the RNA target secondary structure(s) predicted using an energy minimization algorithm (*e.g.*, *mfold*, *RNAstructure*). Moreover, the program disregards any hits that present a bulge loop on either side of the double strand. In other words, both strands are composed of only consecutively paired nucleotides. The input file is the "ct" output file from the *mfold*, *RNAstructure*, or *RNAFold* program [46], which is a common text file format for writing nucleic acid secondary structure. The *TFOFinder* output file lists the most 5' number for the position of the duplex regions identified in the RNA target structure, parallel pyrimidine probe sequence for a user-defined length between 4 and 30 nucleotides and melting temperature for an intermolecular duplex between the TFO RNA and the corresponding complementary RNA sequence (Fig 2B). A target region is identified as a hit if it is predicted to form a R:Y (including G-U pairs) uninterrupted duplex when considering base pairing in all predicted secondary structures for the RNA target of interest [*i.e.*, minimum free energy (MFE) and suboptimal structures (SO)]. When SO structures are included in the "ct" input file, a nucleotide will be considered as double-stranded if it has a corresponding pairing nucleotide in at least one of the structures.

### TFOs for *D. melanogaster* RNA targets

We previously found that it is beneficial to take into consideration predicted suboptimal structures when designing molecular beacon probes for live cell imaging [47]. However, computational time significantly increases when applying minimization algorithms to folding long RNA targets (>11,000-nt), and a dynamic programming algorithm has been shown to not only produce the MFE structure much faster, but also with improved accuracy for long RNA

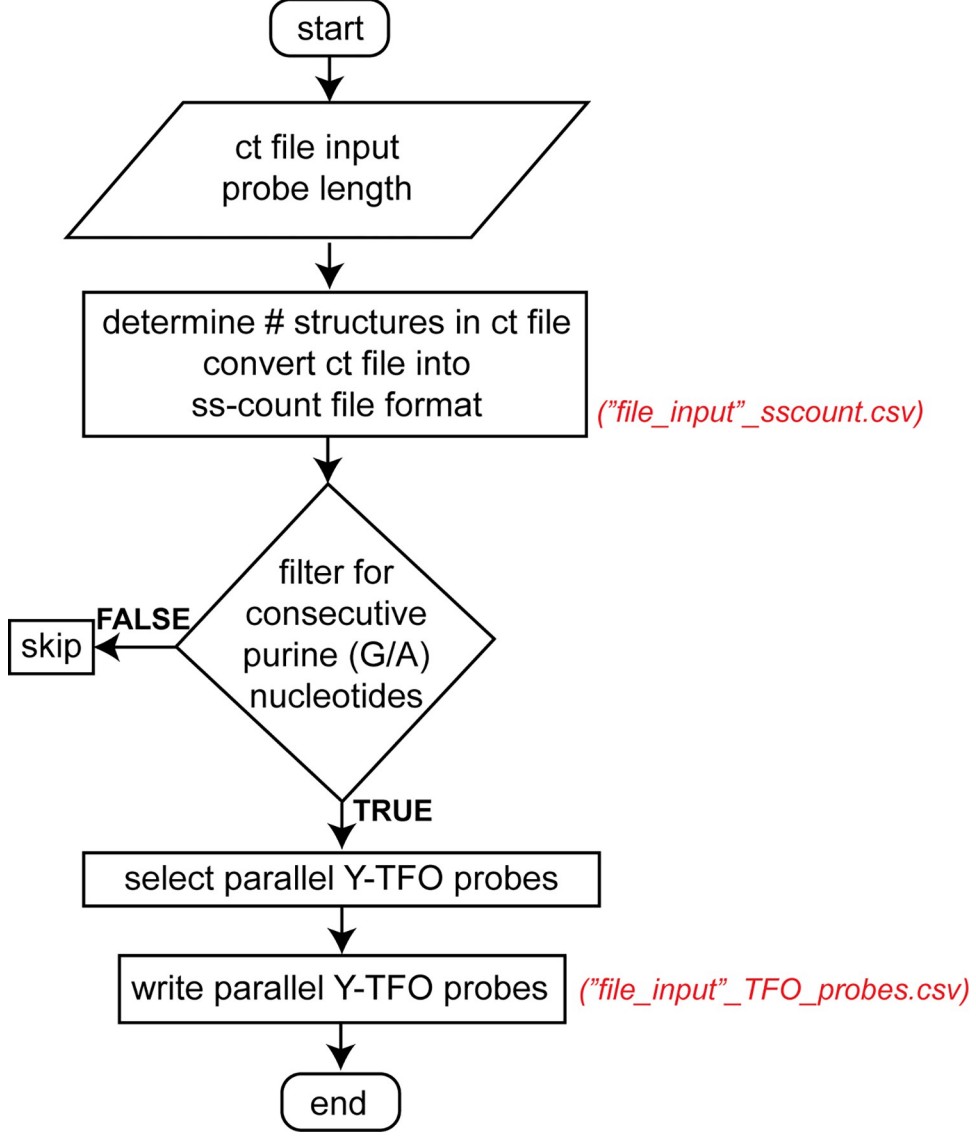

**Fig 4.** *TFOFinder* **program flowchart.**

targets [48]. We used *mfold*, *RNAstructure*, and *LinearFold* to predict the secondary structure of the 123 unique transcripts identified in our *RNAMotif* search, and we analyzed the distribution of the 494 total TTS hits (Table 2, R12 –total double-stranded hits) between the MFE and SO structures (Table 3). We found that 21% (26) of the targets identified using *RNAMotif* did not present a predicted 12-bp duplex amenable to forming an Y•R:Y triplex within their secondary structure(s), while for 19% (23) RNA transcripts the SO structures presented TTS, but the MFE structure did not. The MFE structure of the remaining 60% (74) of transcripts presented at least one TTS.

For TFO targeting to work as intended, probe specificity and sensitivity are essential characteristics. We analyzed the specificity of the TFO probes identified for the 123 transcripts by analyzing all TTS sequences and, of the 4,095 possible unique TTS R12 sequences, 50 were found in the 494 total R12-double-stranded hits (Table 4), with two R12 sequences composed

**Table 3. Distribution of the 494 TTS hits identified in 123 unique transcripts between the MFE and SO structures, which were predicted using minimization algorithms (*mfold, RNAstructure, LinearFold*).**

| MFE TTS | SOs TTS | # Transcripts |
|---|---|---|
| none | none | 14 |
| none | N/A | 12 |
| x | x | 27 |
| none | > 1 | 23 |
| x | y > x | 33 |
| > 1 | N/A | 14 |
| **Total # transcripts** | | **123** |

x, y > 0 are the number of transcripts with purine-only TTS in the MFE and/or SO structures.

of consecutive "GA" or "AG" representing 45% of total hits (223 of 494; Table 4), and contained within 13 unique transcripts mapped to three unique genes (*eag*, *RSG7*, and *CG42260*; S1 Table). Further analysis of these TTS sequences showed that 48% (21 of 50; Table 4) of the identified TTS were unique sequence hits and were mapped to 12 unique transcripts encoded within 12 unique genes.

We sorted the 123 transcripts according to their length, and the first two transcripts were two noncoding RNAs (*CR44598-RA*, 486-nt and *CR44619-RA*, 1,023-nt). For the first one, one TTS was identified when using all target structures (MFE and 13 SO structures; 5' location = 246, ss-count fraction = 0.68 [47]) (Fig 5A), while for the second one, five non-redundant TTS were identified when including the suboptimal structures, but none were found in the MFE structure to have all double-stranded purines. For example, the TTS mapped between 882–894 was present as fully double-stranded, but with one 1-nt bulge on the 3' strand in two (SO# 3, 4) of 19 total structures (Fig 5B, red arrowhead), while in the MFE and ten SO (SO# 1, 2, 11–18) structures, this region presented a mispair (MP) and an 1-nt bulge (Fig 5B, red arrows). The remaining six SO (SO# 5–10) structures presented at least four single-stranded purines. In addition, the ss-count fraction for a TFO probe should be as close to zero as possible, as an ss-count fraction equal to zero means that all TTS nucleotides are base-paired in all structures. The ss-count fraction indicates the extent to which a sequence is predicted to be single-stranded in the MFE and/or SO structures. The larger the value of the ss-count fraction, the more likely it will be that the sequence will have a single-stranded character, where 1 = fully single-stranded and 0 = fully double-stranded. The ss-count fraction was calculated by dividing the sum of the ss-count numbers of the individual bases in the TTS by the product of the probe length and number of total structures (MFE and SO structures) in the input file. The ss-count number represents the number of structures of the total structures in which a base is predicted to be single-stranded, and the ss-count file is one of the output files obtained when predicting RNA secondary structure using *mfold*.

**Table 4. Distribution of the *TFOFinder* identified TTS sequences.**

| # TTS occurrences | # unique TTS | # unique transcripts | # unique genes | # hits |
|---|---|---|---|---|
| ≥ 100 | 2 | 13 | 3 | 223 |
| ≥ 10 | 11 | 93 | 38 | 192 |
| ≥ 2 | 16 | 39 | 16 | 58 |
| 1 | 21 | 12 | 12 | 21 |
| **Total number of TTS hits** | | | | **494** |

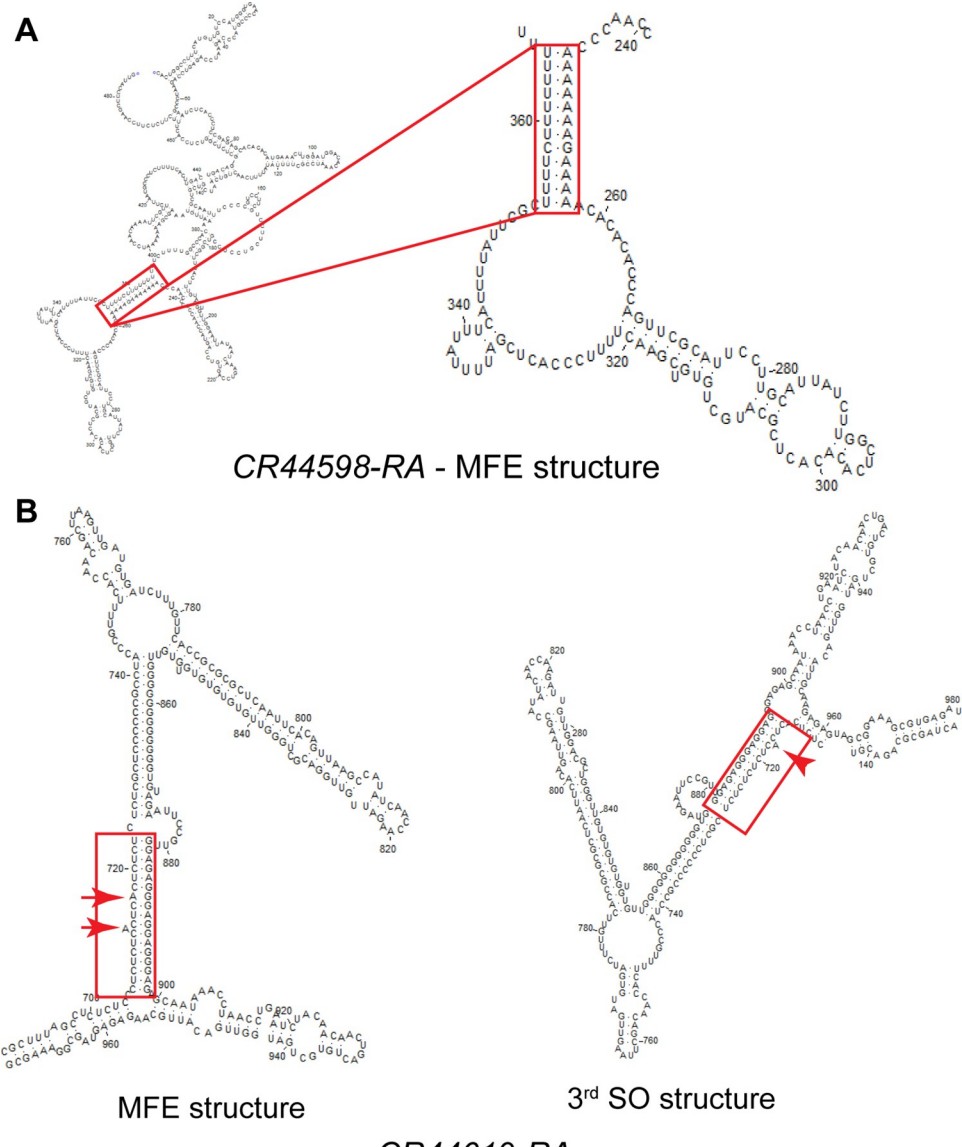

**Fig 5. Secondary structures for two ncRNAs, predicted with *mfold*.** (A) Full MFE structure of the shortest transcript (*CR44598-RA*) identified to contain one TTS, which is highlighted in the red box and shown magnified (right). (B) A longer ncRNA (*CR44619-RA*, 1,023-nt) containing several TTS; one TTS that contains a mispair and one 1-nt bulge in the MFE structure (left, red arrows) is highlighted in the red boxes for the MFE and the 3rd SO structure, in which it presents only one 1-nt bulge (right, red arrowhead).

## TFOs for *Influenza A* vRNA8 target

Using *TFOFinder*, we explored a previously reported RNA target that was shown to form PNA•RNA:RNA triplexes *in vivo* [27]. Partially complementary sequences at the 5' and 3' end of all eight vRNAs of IAV make up a conserved panhandle motif that acts as a viral promoter for transcription and replication. However, this motif contains at least one bulge and therefore it does not fit the ideal requirements for parallel Y·R:Y triple helix formation and requires TFO modification to form a triplex. The panhandle region of vRNA8 was identified as a TFO-target and it was reported that a modified PNA TFO efficiently inhibits IAV replication [27]. Using

the *Clustal 1.2.4* web server [49], we performed a sequence alignment of 15 vRNA8 Viet Nam strains and found that the reported TTS was not conserved among these sequences, which means that the identified TFO would work only for the HM006763A strain (Fig 6, red box).

Therefore, using *TFOFinder*, we searched for additional TTS in the same 15 vRNA8 IAV Viet Nam sequences and compared our results with the experimentally probed secondary

```
CLUSTAL O(1.2.4) multiple sequence alignment

KP638548A/Vietnam/UT36236/20102010/02/208(NS)      --------------TGTT-----------TTTTATCATTAAATAAGCTGAAACGAGAAAG 35
KP638549A/Vietnam/UT36282/20102010/03/278(NS)      --------------TGTT-----------TTTTATCATTAAATAAGCTGAAACGAGAAAG 35
KF277186A/VietNam/CM32/20112011/10/198(NS)         ----GAGATAAGAGCCTTCTCGTTTCAGCTTATTTAATTAAATAAGCTGAAACGAGAAGG 56
AY818147A/VietNam/1203/20042004//8(NS)             ---------------TT-----------TTTTATCATTAAATAAGCTGAAACGAGAAGG 33
HM006763A/VietNam/1203/20042004//8(NS)             AGTAGAAACAAGGGTGTT-----------TTTTATCATTAAATAAGCTGAAACGAGAAGG 49
KP638546A/Vietnam/UT36250-1/20102010/03/058(NS)    --------------TGTT-----------TTTTATTATTAAATAAGCTGAAACGAGAAGG 35
HM114621A/Vietnam/HN31432M/20082008/02/218(NS)     -------------------------------------TTAAATAAGCTGAAACGAGAAGG 23
KP638545A/Vietnam/UT31641-2/20082008/12/198(NS)    --------------TGTT-----------TTTTATCATTAAATAAGCTGAAACGAGAAGG 35
HM114605A/Vietnam/UT31412II/20082008/02/098(NS)    -------------------------------------TTAAATAAGCTGAAACGAGAAGG 23
HM114549A/Vietnam/UT31203A/20072007/05/198(NS)     ------------------------------------CTAAATAAGCTGAAACGAGAAGG 23
HM114613A/Vietnam/UT31413II/20082008/02/138(NS)    -------------------------------------TTAAATAAGCTGAAACGAGAAGG 23
HM114589A/Vietnam/HN31388M1/20072007/12/148(NS)    -------------------------------------TTAAATAAGCTGAAACGAGAAGG 23
HM114557A/Vietnam/UT31239/20072007//8(NS)          -------------------------------------TTAAATAAGCTGAAACGAGAAGG 23
HM114565A/Vietnam/UT31244II/20072007/06/098(NS)    -------------------------------------TTAAATAAGCTGAAACGAGAAGG 23
HM114573A/Vietnam/UT31244III/20072007/06/198(NS)   -------------------------------------TTAAATAAGCTGAAACGAGAAGG 23
                                                                                        ****************** *

KP638548A/Vietnam/UT36236/20102010/02/208(NS)      CTTCAAACTTCTGACTCAATTGTTCTCGCCATTTTCCGTTTCTGATTTGGAGGGAGTGGA 215
KP638549A/Vietnam/UT36282/20102010/03/278(NS)      CTTCAAACTTCTGACTCAATTGTTCTCGCCATTTTCCGTTTCTGATTTGGAGGGAGTGGA 215
KF277186A/VietNam/CM32/20112011/10/198(NS)         CTTCAAACTTCTGACTCAATTGCTCTCGCCAGTTTCCGTTTCTGATTTGGAGGGAGTGGA 236
AY818147A/VietNam/1203/20042004//8(NS)             CTTCAAACTTCTGACTCAATTGTTCTCGCCATTTACCGTTTCTGATTTGGAGGGAGTGGA 213
HM006763A/VietNam/1203/20042004//8(NS)             CTTCAAACTTCTGACTCAATTGTTCTCGCCATTTACCGTTTCTGATTTGGAGGGAGTGGA 229
KP638546A/Vietnam/UT36250-1/20102010/03/058(NS)    CTTCAAACTTCTGACTCAATTGTTCTCGCCATTTACCGTTTCTGATTTGGAGGGAGTAGA 215
HM114621A/Vietnam/HN31432M/20082008/02/218(NS)     CTTCAAACTTCTGGCTCAATTGTTCTCGCCATTTACCGTTTCTGATTTGGAGGGAGTAGA 203
KP638545A/Vietnam/UT31641-2/20082008/12/198(NS)    CTTCAAACTTCTGACCCAATTGTTCTCGCCATTTTCCGTTTCTGATTTGGAGGGAGTGGA 215
HM114605A/Vietnam/UT31412II/20082008/02/098(NS)    CTTCAAACTTCTGACCCAATTGTTCTCGCCATTTTCCGTTTCTGATTTGGAGGGAGTGGA 203
HM114549A/Vietnam/UT31203A/20072007/05/198(NS)     CTTCAAACTTCTGACCCAATTGTTCTCGCCATTTTCCGTTTCTGATTTGGAGGGAGTGGA 203
HM114613A/Vietnam/UT31413II/20082008/02/138(NS)    CTTCAAACTTCTGACCCAATTGTTCTCGCCATTTTCCGTTTCTGATTTGGAGGGAGTGGA 203
HM114589A/Vietnam/HN31388M1/20072007/12/148(NS)    CTTCAAACTTCTGACCCAATTGTTCTCGCCATTTTCCGTTTCTGATTTGGAGGGAGTGGA 203
HM114557A/Vietnam/UT31239/20072007//8(NS)          CTTCAAACTTCTGACCCAATTGTTCTCGCCATTTTCCGTTTCTGATTTGGAGGGAGTGGA 203
HM114565A/Vietnam/UT31244II/20072007/06/098(NS)    CTTCAAACTTCTGACCCAATTGTTCTCGCCATTTTCCGTTTCTGATTTGGAGGGAGTGGA 203
HM114573A/Vietnam/UT31244III/20072007/06/198(NS)   CTTCAAACTTCTGACCCAATTGTTCTCGCCATTTTCCGTTTCTGATTTGGAGGGAGTGGA 203
                                                   ************* * ****** ******** ** ****************** **

KP638548A/Vietnam/UT36236/20102010/02/208(NS)      TCACCAGTATGTCCTGGAAGAGAAGGTAATGGTGAGATCTCTCCCACGATTGCTCCTTCT 395
KP638549A/Vietnam/UT36282/20102010/03/278(NS)      TCACCAGTATGTCCTGGAAGAGAAGGTAATGGTGAGATCTCTCCCACGATTGCTCCTTCT 395
KF277186A/VietNam/CM32/20112011/10/198(NS)         TCACCAGTATGTCCTGGAAGAGAAGGTAATGGTGAGATTCTCCCACGATTGCTCCTTTT 416
AY818147A/VietNam/1203/20042004//8(NS)             TCACCAGTATGTCCTGGAAGAGAAGGTAATGGTGAGATTCTCCCACGATTGCTCCTTCT 393
HM006763A/VietNam/1203/20042004//8(NS)             TCACCAGTATGTCCTGGAAGAGAAGGTAATGGTGAGATTCTCCCACGATTGCTCCTTCT 409
KP638546A/Vietnam/UT36250-1/20102010/03/058(NS)    TCACCAGTATGTCCTGGAAGAGAAGGTAATGGTGAGATTCTCCCACGATTGCTCCTTCC 395
HM114621A/Vietnam/HN31432M/20082008/02/218(NS)     TCACCAGTATGTCCTGGAAGAGAAGGTAATGGTGAGATTCTCCCACGATTGCTCCTTCT 383
KP638545A/Vietnam/UT31641-2/20082008/12/198(NS)    TCACCAGTATGTCCTGGAAGAGAAGGTAATGGTGAGATCTCTCCCACAATTGCTCCTTCT 395
HM114605A/Vietnam/UT31412II/20082008/02/098(NS)    TCACCAGTATGTCCTGGAAGAGAAGGTAATGGTGAGATCTCTCCCACGATTGCTCCTTCT 383
HM114549A/Vietnam/UT31203A/20072007/05/198(NS)     TCACCAGTATGTCCTGGAAGAGAAGGTAATGGTGAGATCTCTCCCACGATTGCTTCTTCT 383
HM114613A/Vietnam/UT31413II/20082008/02/138(NS)    TCACCAGTATGTCCTGGAAGAGAAGGTAATGGTGAGATCTCTCCCACGATTGCTCCTTCT 383
HM114589A/Vietnam/HN31388M1/20072007/12/148(NS)    TCACCAGTATGTCCTGGAAGAGAAGGTAATGGTGAGATCTCTCCCACGATTGCTCCTTCT 383
HM114557A/Vietnam/UT31239/20072007//8(NS)          TCACCAGTATGTCCTGGAAGAGAAGGTAATGGTGAGATCTCTCCCACGATTGCTCCTTCT 383
HM114565A/Vietnam/UT31244II/20072007/06/098(NS)    TCACCAGTATGTCCTGGAAGAGAAGGTAATGGTGAGATCTCTCCCACGATTGCTCCTTCT 383
HM114573A/Vietnam/UT31244III/20072007/06/198(NS)   TCACCAGTATGTCCTGGAAGAGAAGGTAATGGTGAGATCTCTCCCACGATTGCTCCTTCT 383
                                                   *******************************************  ******** ****** ***
```

**Fig 6. Alignment of 15 vRNA8 IAV sequences (*Clustal 1.2.4* web server).** The red box highlights the panhandle TTS experimentally targeted for inhibiting influenza A replication. The green boxes highlight two conserved TTS identified using *TFOFinder*.

**Table 5.** *TFOFinder* **results for vRNA8 IAV Viet Nam strain HM006763.**

| Length | 5'Target no. for HM006763A | Predicted MFE | | Experimental | |
|---|---|---|---|---|---|
| 11 | 365 | GGA_AGAGAaGG | bulge | GGaAGAGAaGG | multibranch loop |
| 8 | 218 | GGA_GGGAG | 1-nt bulge | GGA_GGGAG | 1-nt bulge |
| 8 | 804 | AaaGAAAG | 2x1 internal loop | AaaGAAAG | 2x1 internal loop |

Small letters = single-stranded base; underscore = 3' strand bulge

structure of the target vRNA [40]. When including in the search the MFE and SO structures, we identified three conserved regions, two of which are highlighted in Fig 6 (green boxes, TTS positioned at 365 and 218) (Table 5). This means that the MFE structure did not present an ideal TTS, but each purine contained in the identified TTS was double-stranded in at least one of the SO structures. The first and third TTS (Table 5: TTS positioned at 365 and 804) do not appear to be good candidates to form a triplex because the former is part of a multibranch loop, and the latter includes an internal loop [40]. However, the reported structure was determined using solution assays and it is possible that the *in vitro* structure may differ from the *in vivo* folding of the RNA target, although one would expect the *in vivo* folding to be less structured [50]. The second TTS (Table 5, TTS positioned at 218) may be a viable alternative and is conserved in all strains, but it is shorter than the recommended minimum length (8 vs. 10-nt), which may compromise the sensitivity and specificity of the assay for the targeted TTS. To assess the specificity of this probe, using *RNAMotif*, we performed for the IAV TTS-218 similar searches as described for the *D. melanogaster* transcriptome for both *D. melanogaster* (*version 6.38*) and *H. sapiens* (May 23[rd], 2018) transcriptomes (Table 6). We found that in *D. melanogaster*, only 0.04% of transcripts had the potential to form the double-stranded IAV TTS-218, while in *H. sapiens* this percentage increased to 2.58%, which was still small. However, a longer TTS would make a more attractive region to design modified TFOs for functional inhibition.

## Conclusion

*TFOFinder* is a platform-independent Python program for the fast and efficient identification within any RNA structure of purine-only double-stranded regions that are predicted to form parallel triple helices of the TFO•RNA:RNA type. The design of target-specific TFO probes is applicable to studies of *in vivo* RNA structure, RNA imaging, and RNA function regulation.

## Materials and methods

### Target sequences

**D. melanogaster transcriptome and genome.** For the survey of *Drosophila melanogaster* targets, the corresponding FASTA sequences were downloaded using the *Flybase* online tools

**Table 6.** *RNAMotif* **results for IAV TTS-218 prevalence in** *D. melanogaster* **and** *H. sapiens* **transcriptome.**

| Organism | Single-stranded IAV218 TTS | | | Double-stranded IAV218 TTS | | |
|---|---|---|---|---|---|---|
| | total | # Unique transcripts | % Unique transcripts | total | # Unique transcripts | % Unique transcripts |
| *D. melanogaster* | 1,270 | 1,254 | 3.52 | 15 | 15 | 0.04 |
| *H. sapiens* | 15,199 | 11,720 | 15.51 | 3,435 | 1,950 | 2.58 |

Total number of transcripts = 35,642 and 75,573 for *D. melanogaster* and *H. sapiens*, respectively.

[51]. The full transcriptome *version 6.38 (02/18/2021)* and genome *version 6.48 (09/26/2022)* were used to perform the surveys.

**Influenza A vRNA8.**    The full-length segment 8 sequences of IAV Viet Nam strain were downloaded from the NCBI (National Center for Biotechnology Information) Influenza Virus Resource [52]. The reverse complement of these 15 sequences, which are the vRNA sequences, were generated using *BioEdit* [53], folded using *Fold-smp* from the *RNAstructure version 6.4* [54] using the previously reported SHAPE data file and constraints (slope = 2.6 and intercept = -0.8) [40]. The resulting "ct" files, which contained information about the secondary structure of the MFE and up to 19 SO structures, were used to identify TFO-target regions using a batch version of *TFOFinder*.

*Homo sapiens* **refseq_rna.**    The FASTA sequences were downloaded from the NCBI download site last updated on May 23rd, 2018, using the Aspera download tool [NCBI>refseq>H_sapiens>mRNA_protein>human.X.rna.fna.gz, (X = 1, 2, 10, 11, and 12)].

*D. melanogaster* **genome survey for purine-rich sequences.**    We searched for 12 consecutive purines, including all As [R(A)12] on both strands of the *D. melanogaster* DNA sequences downloaded in FASTA format as gene, mRNA, ncRNA, miRNA, tRNA, exon, intron, intergenic, 5'UTR, and 3'UTR. The transcriptome search described below identified single-stranded purine sequences that corresponded to the double-stranded DNA encoding each transcript. However, the transcriptome survey did not consider the intergenic and intronic parts of the DNA genome. Moreover, many of the hits found in transcripts were redundant as many genes encode for several mRNA variants with overlapping sequences.

*D. melanogaster* **transcriptome survey for purine-rich sequences.**    We identified purine-rich sequences in all *D. melanogaster* transcripts by performing one-strand searches using *RNAMotif*. Several examples of descriptor files used with *RNAMotif* can be found in the supporting information section (S1 Text). We searched for single-stranded purine-only sequences composed of consecutive purines (R12) that were not all adenines (A), or contained only As (A12), or for purine-rich regions interrupted by up to two pyrimidines (R11Y and R10Y2). We next searched within the identified transcript sequences for complementary regions that can form a duplex with the already identified single-stranded hits. To confirm our results, we also performed this search on the full transcriptome, and the two searches yielded the same hits.

**Identification of transcripts with single-stranded purine-rich stretches.**    Using *RNAMotif*, we scanned the transcriptome of *D. melanogaster* for stretches of 12 purines, all adenine (Table 2, A12, single-stranded), containing at least one guanine (Table 2, R12, single-stranded), or up to two pyrimidines (Table 2, R11Y, R10Y2 strict and relaxed, single-stranded). From the *RNAMotif* output, we extracted all unique transcript IDs and downloaded their sequences in FASTA format using the *FlyBase Sequence Downloader* tool.

**Identification of transcripts with double-stranded purine-rich stretches.**    Using *RNAMotif*, we then identified transcripts containing the corresponding complementary pyrimidine sequence(s) (Table 2, A12, R12, double-stranded). From the *RNAMotif* output file we extracted the transcript name, length, and genomic location, and the corresponding IDs were downloaded using the *FlyBase Batch Download* tool. The search was then relaxed to allow for G-U pairs (Table 2, R12_GU, double-stranded), or one mispair (Table 2, R12_1MP, double-stranded; Fig 2, 5'R12_1MP), or for one (Table 2, R11Y, double-stranded; Fig 2, 5'R11Y), or two pyrimidine inversions either anywhere in the 12 sequence (Table 2, R10Y2 relaxed, double-stranded) or restricted to the 10 internal positions and not consecutive (Table 2, R10Y2 strict, double-stranded). After identifying all TTS showing the potential to be double-stranded, we predicted the secondary structure(s) of the transcripts that contained them using a minimization algorithm. Using *TFOFinder* we analyzed the likelihood of each TTS to be double-

stranded in the predicted secondary structure(s). To find the predicted MFE secondary structure of the transcript, we used *LinearFold* for RNA targets longer than 11,000-nt, *mfold* for transcripts with up to 2,400-nt, and *RNAstructure* for the remaining sequences. In addition to the MFE structure, *RNAstructure* and *mfold* provided a various number of suboptimal structures. Using *TFOFinder*, we took into consideration the predicted secondary structure(s) to identify regions of 12 double-stranded purines.

**Analysis of hits.** Using gawk, custom Python scripts, and *Flybase* tools, we extracted the ID of the unique transcripts and the corresponding unique genes to which the hits were mapped.

### *TFOFinder* program

The open-source program was written in Python with a text interface, and it is freely available on GitHub (https://github.com/icatrina/TFOFinder). The input file is the "ct" format file, which is used to count the total number of structures (MFE and SO), identify consecutive purines of a user-defined length (4-30-nt) and list in the output file information for the parallel (5' ➜ 3') TFO probes forming Y·R:Y triplexes. The output lists the 5' start position for the identified TTS that can form a Y·R:Y parallel triplex, the percentage of G/A content of the RNA TTS, the parallel TFO sequence, and the melting temperature ($T_m$) of the duplex of the RNA TFO and the corresponding complementary RNA sequence. Alternatively, the *TFOFinder* can be used via free Amazon Web Services (AWS), with AWS CloudShell, which allows for up to 1GB free persistent storage.

A tutorial file can be found in the above-mentioned GitHub repository. This tutorial provides details for the download and installation requirements, as well as the usage of *TFOFinder* for the 67[th] RNA target, *ovo-RE* mRNA (S1 Table). The input and output files for this example are also provided.

### Supporting information

**S1 Table D. *melanogaster* unique transcripts with the potential of forming at least one R12 double-stranded region, identified using *RNAMotif*.**
(XLSX)

**S1 Text. Example of descriptors used for the *RNAMotif* searches.**
(PDF)

### Acknowledgments

We are very thankful and grateful to Dave Matthews, M.D., Ph.D. (University of Rochester) for his continuing support with *RNAstructure* algorithms and for his invaluable help and advice on thermodynamic analysis of nucleic acid folding, programming, and more. We thank Livia V. Bayer, Ph.D., (Hunter College, CUNY) for critically reading this manuscript and helpful discussions. We are also grateful to the *Flybase* help team, and in particular to Josh Goodman, Julie Agapite, and Victor B. Strelets, for promptly answering our questions and writing customized scripts to meet our needs. Finally, we would like to thank current and past members of the Catrina laboratory for their experimental work that has contributed to the planning of this computational analysis.

### Author Contributions

**Conceptualization:** Irina E. Catrina.

**Data curation:** Atara Neugroschl, Irina E. Catrina.

**Formal analysis:** Atara Neugroschl, Irina E. Catrina.

**Funding acquisition:** Irina E. Catrina.

**Investigation:** Atara Neugroschl, Irina E. Catrina.

**Methodology:** Irina E. Catrina.

**Project administration:** Irina E. Catrina.

**Resources:** Atara Neugroschl, Irina E. Catrina.

**Software:** Atara Neugroschl, Irina E. Catrina.

**Supervision:** Irina E. Catrina.

**Validation:** Atara Neugroschl, Irina E. Catrina.

**Visualization:** Atara Neugroschl, Irina E. Catrina.

**Writing – original draft:** Atara Neugroschl, Irina E. Catrina.

**Writing – review & editing:** Atara Neugroschl, Irina E. Catrina.

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
