## [Decision Letter · Decision Letter 0]

17 Jun 2023

Dear Catrina,

Thank you very much for submitting your manuscript "TFOFinder: Python program for identifying purine-only double-stranded stretches in the predicted secondary structure(s) of RNA targets" for consideration at PLOS Computational Biology. As with all papers reviewed by the journal, your manuscript was reviewed by members of the editorial board and by several independent reviewers. The reviewers appreciated the attention to an important topic. Based on the reviews, we are likely to accept this manuscript for publication, providing that you modify the manuscript according to the review recommendations.

Sincerely,

Alexander MacKerell

Academic Editor

PLOS Computational Biology

William Noble

Section Editor

PLOS Computational Biology

Reviewer's Responses to Questions

**Comments to the Authors:**

Reviewer #1: Neugroschl and Catrina describe TFOFinder, their Python program that identifies intramolecular purine-only RNA duplexes that are amenable to forming parallel triple helices. They used this program to analyze the Drosophila genome for potential triplex target sites. The manuscript appears to be well-written. In my opinion, while the tool may be helpful for a small group of scientists interested in targeting triplex sites, this appears to be a simple, incremental step that won’t have much impact on the field. The authors may want to consider the following:

• Perhaps an explanation of how RNAMotif and TFOFinder differ, how they can be used in conjunction, and how TFOFinder advances the field would be beneficial for the readership.

• Line 67, “therapeutics approaches” should be “therapeutic approaches.”

• Line 77, what is meant by “frame”? Backbone?

• Line 81, “in a greater mismatch discrimination” should be “with a greater mismatch discrimination.”

• Line 100, “duple-formation” should be “duplex-formation.”

• Line 134, FRET should be defined.

• Line 133, this paragraph seems to be out of place. The authors provided introductory material and then introduced their new tool. This paragraph of introductory material sits between two paragraphs that discuss the tool. Is there a better location in the introduction to move this paragraph?

• Line 259, MFE and SO are used but not defined until lines 374-375.

• Line 298, what is sscount?

• Line 309, “one highlighted in the red box for the of the 12th 310 SO structure” does not make sense and needs editing.

• Line 377, define NCBI.

• Ref 14 doesn’t look complete.

Reviewer #2: General Comment

The authors have presented a potentially valuable and innovative tool for designing TFOs targeting RNA in the model species D. melanogaster (genome and transcriptome) and the vRNA8 of influenza A. They have searched for double-stranded fragments of a user-defined length (4-30 nt) composed of consecutive purines within predicted secondary structures of the RNA target of interest.

The literature review and description of the methods employed by the authors are clear and concise, and the rationale for the study is evident. We appreciate the authors providing the link to the Github repository containing the TFOFinder python code. While we believe that the wider scientific and bioinformatics community can benefit from this work, we suggest the authors consider applying the FAIR (Findable, Accessible, Interoperable, and Reusable) principles to the manuscript to ensure reproducibility and reusability of the codebase. It would be helpful if the authors could provide test data to demonstrate the usage of the provided scripts and how they integrate with other tools used in the complete study. Additionally, clearer documentation of the code is necessary to further enhance the credibility of the study.

The application of TFOFinder for identifying conserved TTS in the influenza A virus was a notable achievement. Based on their analysis, we wonder if the authors are considering conducting further studies on RNA targets of other respiratory viruses or perhaps viruses that affect plants?

**Have the authors made all data and (if applicable) computational code underlying the findings in their manuscript fully available?**

Reviewer #1: Yes

Reviewer #2: Yes

PLOS authors have the option to publish the peer review history of their article (what does this mean?). If published, this will include your full peer review and any attached files.

Reviewer #1: No

Reviewer #2: No

Figure Files:

Data Requirements:

Reproducibility:

References:

---

## [Editor Report · Decision Letter 1]

8 Aug 2023

Dear Catrina,

We are pleased to inform you that your manuscript 'TFOFinder: Python program for identifying purine-only double-stranded stretches in the predicted secondary structure(s) of RNA targets' has been provisionally accepted for publication in PLOS Computational Biology.

Best regards,

Alexander MacKerell

Academic Editor

PLOS Computational Biology

William Noble

Section Editor

PLOS Computational Biology

---

## [Editor Report · Acceptance letter]

22 Aug 2023

PCOMPBIOL-D-23-00670R1 

TFOFinder: Python program for identifying purine-only double-stranded stretches in the predicted secondary structure(s) of RNA targets

Dear Dr Catrina,

I am pleased to inform you that your manuscript has been formally accepted for publication in PLOS Computational Biology. Your manuscript is now with our production department and you will be notified of the publication date in due course.

With kind regards,

Anita Estes
